# GENERATING SYNTHETIC GENOTYPES USING DIFFUSION MODELS

## ABSTRACT

In this paper, we introduce the first diffusion model designed to generate *complete* synthetic human genotypes, which, by standard protocols, one can straightforwardly expand into full-length, DNA-level genomes. The synthetic genotypes mimic real human genotypes without just reproducing known genotypes, in terms of approved metrics. When training biomedically relevant classifiers with synthetic genotypes, accuracy is near-identical to the accuracy achieved when training classifiers with real data. We further demonstrate that augmenting small amounts of real with synthetically generated genotypes drastically improves performance rates. This addresses a significant challenge in translational human genetics: real human genotypes, although emerging in large volumes from genome wide association studies, are sensitive private data, which limits their public availability. Therefore, the integration of additional, insensitive data when striving for rapid sharing of biomedical knowledge of public interest appears imperative.

## 1 INTRODUCTION

Deep learning has enabled significant advancements in the field of computational biology (Jumper et al., 2021; Cheng et al., 2023; Wong et al., 2024). However, the vast majority of such approaches resort to processing smaller portions of human genomes, such as coding regions and their products (proteins), or small local segments of human genomes (Guo et al., 2023; Shen et al., 2022).

The reasons for this are threefold. First, the enormous length of human genomes prevents their straightforward usage in neural network architectures. Second, whole human genome data are expensive, because of the massive clinical and experimental efforts required in order to obtain them. Third, whole human genome data are usually subject to strict access regulations because of privacy concerns. The difficulty to process, gather and share sufficient (training) data impedes scientific progress through reliable extraction of knowledge, rapid dissemination of relevant data, and render easy and full reproducibility of already obtained results impossible.

While the first reason is a (challenging) technical concern, the second and the third reason establish our key motivation from the point of view of applications in biomedicine.

The solution that we suggest here is the generation of synthetically generated whole genome data that is cheap, easy to gather, and privacy-enhancing. We present a diffusion model based framework that can generate synthetic whole-genome human genotype data.

Despite the relatively small amounts of data used during training, we ensure that our diffusion models do, in fact, *generate novel whole genome human genotypes.* By means of approved reliable metrics, we ensure that the synthetically generated genotypes are of high quality, that is realistic in terms of stemming from the distribution governing real human genotypes, and also diverse, that is they do not exactly reproduce the individual genomes used for training the diffusion models, which translates into preservation of privacy in the setting at hand.

Note that, unlike previous work (Guo et al., 2023), whole-genome genotypes can be straightforwardly expanded into whole DNA-level genomes, by means of applicable genome reference systems. Furthermore, diffusion models enable us to generate disease-affected and non-disease-affected genotypes in a targeted manner.

Beyond demonstrating that the synthetically generated genomes are realistic by the measures that ensure that the diffusion model approximately captures the distribution of human genotypes, we further demonstrate that training classifiers (Luo et al., 2023) using synthetically generated data, by either integration or replacement, and evaluating them on the original data achieves performance rates that rival those of the original classifiers. This provides further evidence that the diffusion model has not only captured the general structure of human genomes, but also has picked up the mechanisms that distinguish diseased from non-diseased genotypes.

## 2 RELATED WORK

### 2.1 PREVIOUS MODELS

In Table 1, we systematically compare most previous work which has tried to produce genomes. To the very best of our knowledge, we are the first approach which succeeds in generating full-length human genotypes. After thoroughly and carefully revisiting the landscape of existing tools and approaches (see Table 1), we conclude that there are indeed no approaches that can generate synthetic full-length human genotypes (or even human genomes directly at the level of DNA). All approaches presented so far address generating smaller segments of human genomes, most not even spanning the length of one chromosome. However, classifiers as the one presented in Luo et al. (2023) require full-length human genome data to work well.

Table 1: An overview of related work on generating synthetic genomes and its differences / similarities in comparison with our work. Row headers are: Reference, the modeling approach used, the data type the model works on, length of generated genomes presented, whether or not the model can be conditioned to produce specific types of data. Our novelties are **highlighted**.

| Reference | Model | Data Type | Genome Length | Cond. |
|---|---|---|---|---|
| DNAGPT Zhang et al. (2023) | Autoregressive | Base-Pairs | 24k BPS | x |
| HyenaDNA Nguyen et al. (2023) | Autoregressive | Base-Pairs | $10^6$ BPS | x |
| HAPNEST Wharrie et al. (2023) | LD & Markov | SNPs | 1 Chromosome | x |
| Perera et al. (2022) | GMMNs | SNPs | 1 Chromosome | ✓ |
| Yelmen et al. (2021) | GAN,RBM | SNPs | 10k SNPs | x |
| Yelmen et al. (2023) | WGAN | SNPs | 10k SNPs | x |
| Szatkownik et al. (2024) | WGAN | PCA+SNPs | 65k SNPs | x |
| Ahronoviz & Gronau (2024) | GAN | SNPs | 10k SNPs | ✓ |
| Burnard et al. (2023) | VAE | SNPs | 1 Chromosome | x |
| Dang et al. (2023) | HCLTs | SNPs | 10k SNPs | x |
| GeneticDiffusion (Ours) | **Diffusion** | PCA+SNPs | **Full Genome** | ✓ |

### 2.2 GENERATIVE MODELS

We prefer diffusion models over alternative generative deep learning models for the following reasons: a) Diffusion models have become the tool of choice in generative modeling (Dhariwal & Nichol, 2021), and b) previous successes in the use of diffusion models in regulatory human genomics (Sarkar et al., 2024; Avdeyev et al., 2023; Senan et al., 2024; Li et al., 2024) further support their usage. Unlike these works, we emphasize that in our work we make use of the full length of genotypes, and do not have to restrict ourselves to smaller portions of the human genome. We considered skipping steps for generation as done in DDIM (Song et al., 2022) as a way to speed up generation, but realized that the high quality generated by using the full step length for our generation was important for our use case. Furthermore, we considered classifier free guidance (Ho & Salimans, 2022) to increase the quality of the conditioning during generation, like in Azizi et al. (2023), but observed no positive effects for classification accuracy on the test data.

## 2.3 WORKING WITH LONG SEQUENCES

One of the driving problems when working with genetic data is the enormous length of the genomes. This is exacerbated by the long range interactions that affect parts of the genomes that are far apart in terms of the sequential order in which they appear.

It is therefore no surprise that recent genomics research is employing techniques that can accommodate the large length of human genomes e.g. HyenaDNA (Nguyen et al., 2023), which is a powerful architecture that is able to process up to a million tokens simultaneously via an adapted form of attention, and also, as of most recently, state space models (Schiff et al., 2024), for which similar principles apply. However, even with those recent methodological advances, sequences of billions, and not just millions in length, cannot be processed.

In the domain of diffusion models, works have been presented, for example "Stable Diffusion" (Rombach et al., 2022), that analogously adopt the paradigm of no longer working directly on the raw data, but rather on appropriately embedded versions of it. We adopt this paradigm.

## 3 METHODS

In the following, we describe the data representations that reflect human genomes, i.e. the genotype profiles that correspond to them, and the computation of embeddings for these genotype profiles, as well as the architectural choices for the diffusion model. For more details on diffusion models and the human Genotype, see the Appendix.

### 3.1 DATA

We deal with two data sets of human genotypes:

**ALS Data.** The individual genotype profiles that we use for training (and testing) in the following, were raised in the frame of Project MinE pro (2018), which is concerned with the study of amyotrophic lateral sclerosis (ALS). As a disease, ALS is of particular interest to AI based applications, because ALS is driven by complex, still insufficiently understood mutation patterns that escape the grasp of human-understandable approaches. For exactly these reasons, also earlier studies (Auer et al., 2012; Dolzhenko et al., 2017) focus on ALS, using data gathered through Project MinE.

While Project MinE establishes a data resource that is exemplary in terms of size and comprehensiveness, access to its data is subject to strict safety regulations. This is the reason why we exclusively deal with a Dutch cohort of people, for which we were provided access, while not with cohorts of genotypes referring to other countries. The Dutch cohort we worked with consisted of 3292 individuals affected by ALS and 7213 individuals known not to be affected by ALS, by ancestral relationships.

Accordingly, for this data, labels $y$ used to steer the generation of new samples, refer to individuals affected with ALS and without ALS.

**1000 Genomes (1KG).** We also consider the 2504 individuals sequenced in Stage 3 of the 1000 Genomes project Consortium et al. (2015). For these 2504 individuals, alleles were assigned to ancestors (referred to as "phased" in genetics), such that one obtains two *haplotypes* instead of one genotype for each individual, where a haplotype is a binary-valued vector of length $N$ where 0 reflects that the reference allele applied at the particular position whereas 1 reflects to observe the alternative allele at the respective SNP site. Adding up the two haplotypes, entry by entry, results in the genotype of the individual. Because, unlike genotypes, the haplotypes assign alleles to ancestors, they carry more information. This is considerably more valuable for genetics, because they provide immediate insight into the ancestral relationships affecting genomes.

When dealing with 1KG data, we seek to generate haplotype profiles instead of merely genotype profiles. Unlike with Project MinE, no particulars about the corresponding phenotypes are known in the frame of the 1KG project. The only known phenotype is the population from which they stem. So, the additional conditioning input $y$ refers to these 26 population labels when working with 1KG data.

## 3.2 EMBEDDING GENOTYPES / HAPLOTYPES.

In the following, refer to Figure 1 for an illustration the embedding procedure. See also (Luo et al., 2023) for details on the following.

Embedding refers to turning ternary-valued (genotypes, ALS) or binary-valued (haplotypes, 1KG) vectors of length approximately 3-5 millions into real-valued vectors, whose dimension is in the tens of thousands. This does not only reduce the dimensionality of the data, but also ensures consistency in terms of arranging the data for appropriate processing by the diffusion model.

For that transformation, we consider the genes recorded for the ALS and 1KG datasets, amounting to 18279 and 26624 genes, respectively. Based on approved principles, we assign each SNP site to one of the genes. The number of SNP sites per gene can vary quite substantially. Depending on length and location in the genome, a gene can collect roughly between 5 and 100 SNP sites. In other words, each gene reflects a ternary- (ALS) or binary-valued (1KG) vector of length between 5 and 100. Following the approach described in (Luo et al., 2023), we apply principal component analysis (PCA) to each of the 18279 (ALS) or 26624

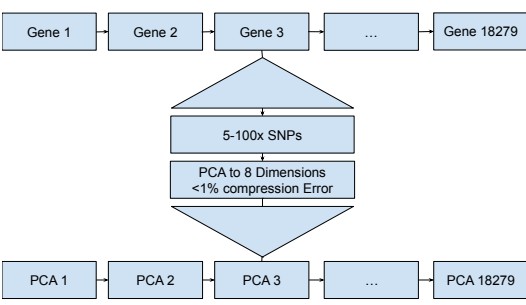

Figure 1: Pre-processing pipeline

binary-valued (1KG) vectors of length between 5 and 100, for each gene separately. This amounts to 18279 (ALS) resp. 26624 (1KG) PCAs, each one applied to the vector segments referring to one of the genes. The result is a collection of principal components for each of the genes, both in the case of ALS and the case of 1KG. Depending on the number of SNP sites per gene, we pick between 1 and 8 principal components (PCs) for each of the 18 279 (ALS) or 26624 (1KG) genes. For consistency and due to architectural reasons, we pad any such vector of length less than 8 (corresponding to less than 8 PCs for the particular gene) with zeros to extend it to length 8.

For both ALS and 1KG, the reduction of dimension comes at minimal compression loss ($< 1\%$) as shown in Luo et al. (2023). This strongly implies that one can decompress the PC embedded data into the original genotypes or haplotypes at only a minor loss of information.

Observing that $18432 = 2^{11} \times 9$, we further pad embeddings $x \in \mathbb{R}^{18279 \times 8}$ with further zeros, extending individual genotype embeddings into elements of $\mathbb{R}^{18432 \times 8}$, which aims at efficiency gains with respect to modern hardware architecture. To remedy the issue that zero padded position add additional failure points all values at padded positions are clamped to zero during training and the generation process. We apply the same procedure for the $26624 \times 8$ - dimensional vectors referring to the 1KG data.

## 3.3 MODEL ARCHITECTURE

We base our Diffusion Model on the popular U-Net Architecture suggested by Ronneberger et al. (2015), while applying modifications that account for the sequential nature of the genetic data. We note that the order of the genes along the genome implies a natural order of the different 8-dimensional gene vectors.

We explore different variants of the basic U-Net architecture by replacing the 2D convolutional layers with their 1D counterparts, or with multi-layer perceptrons (MLPs).

We also explore a transformer encoder structure similar to the one presented in Devlin et al. (2019), but with learnable positional embeddings, and two additional tokens accounting for time steps $t$ and class labels $y$, further equipped with an embedding layer similar to the one used for images in (Dosovitskiy et al., 2020). This serves the purpose of reducing the number of input tokens which is essential for reducing the computational cost. In Figure 2, we visualize the UnetMLP architecture that we propose, it is similar to the convolutional Unet, but does not include multi-headed attention in the intermediate layers. For visualizations of the Transformer and UnetCNN architectures see the Appendix.

In general, we want to point out that the approach using 1D convolutions prioritizes short- to medium-length interactions within the genome, but does only allow for limited long-range interactions. On the other hand, this greatly reduces the amount of parameters to be learned, which offsets the disadvantages from a practical point of view. However not including any kind of spatial information does lead to problems on the presented task, due to the high sensitivity of information with regards towards the position. I.e. it is highly important that the model has positional information about the PCA it processes. This is not the case for the CNN architecture.

In contrast, models solely incorporating dense layers are not subject to sequential biases. However, as in other domains of applications of neural networks, fully connected layers tend to fail to find suitable solutions due to the over-parameterization.

Fully attention based models, for example, models based on transformer encoder architectures do not have any inherent spatial bias, the positional encoding used induces the kind of bias. Following, they should be in theory applicable for this kind of data. Therefore, we also explore such architectures here.

**Combining Models** Since CNN and MLP based models focus on different aspects of the structure of the genome, we suggest combining them into a single network that benefits from the strengths of the two architectural choices, and synthesizes their advantages. We refer to this combination as CNN + MLP. We combine the two separate models $MLP(x, t, y)$ and $CNN(x, t, y)$ during training, and predict the noise (that the diffusion model has added to the input) accordingly:

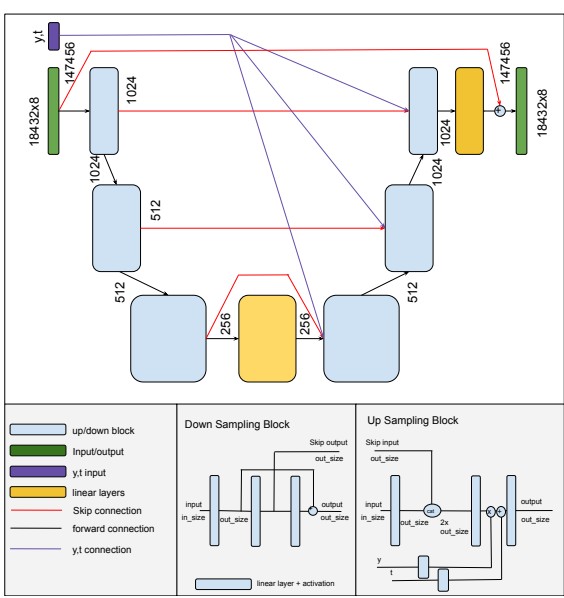

Figure 2: A structural overview of the architecture of the MLP diffusion model.

$$(MLP + CNN)(x, t, y) = (1 - \lambda(t)) \cdot MLP(x, t, y) + \lambda(t) \cdot CNN(x, t, y) \quad (1)$$

where $\lambda(t), t \in [0, 1]$ reflects a learnable function, realized by a straightforward 2-layer-perceptron receiving the noise schedule $t$ as input.

Overall, we explore 4 different diffusion model architectures: "Unet MLP", "Unet CNN", "Unet MLP + CNN" and "Transformer". We perform extensive hyper-parameter tuning on all of these model types and present the best results in the evaluation.

### 3.4 EVALUATION

Evaluation of synthetically generated genomes requires careful consideration. The driving underlying principles are realism, on the one hand, and diversity on the other hand. While realism refers to synthetic genomes being likely to stem from the true distribution of genomes, diversity is concerned with sampled synthetic genomes being sufficiently far away from the true training data points. In image generation common scores to measure these metrics are the Fréchet inception distance (FID) (Heusel et al., 2017) or, the Inception score (Salimans et al., 2016), for example. While human eyesight does not apply in genomics for obvious reasons, the FID and IS cannot be computed either because of the integration of pre-trained large scale networks. In fact, this scenario does not apply in genomics/genetics for exactly the reasons that motivate our work: the lack of available (accessible) large-scale data hampers conventional ML practice. Due to these reasons we are forced to rely on metrics which while proven are a bit more unconventional for the image generation domain namely: Adversarial Accuracy and Classifier Performance.

We note that all datasets we have access to are, although fairly large from the point of view of biomedicine, considerably limited in terms of ML concerns (number of samples at most 10 405). So, one cannot expect the diffusion model to perform at the level of realism observed in other complex domains (such as images and text). This explains why the evaluation relates to exploring the upper limits of possibilities in our context. Note however that our approach virtually serves the purpose of training diffusion models on large-scale data, as hosted by large, access restricted databases, in a safe, access-restricted enviroments. These databases could then provide safe, privacy-preserving large-scale synthetic data on demand, by drawing samples from the diffusion model, without having to publish neither real data nor the diffusion model trained on real data.

### 3.4.1 RECOVERY RATE

Employing synthetic data for training reflects using samples from a distribution that was estimated using empirical data. Since all knowledge captured by that distribution stems from the real, empirical data from which it was estimated, any samples drawn from that distribution can, at most, convey the knowledge that contributed to its estimation. In terms of classifier performance, this means that performance rates achieved when using real data for training establish an upper bound for the performance rates that one can achieve when using synthetic data which was generated by observing the same real data for training.

Somewhat more formally, consider a classifier $C$ and a generator $G$, both of which are trained on the same real data $D_r$. While $G$ explicitly addresses to approximate the distribution that governs $D_r$, $C$ implicitly approximates it in order to establish sufficiently accurate classification boundaries. Sampling synthetic data corresponds to drawing data $D_s$ from the distribution approximated by $G$. So, using $D_s$, the synthetic data, instead of $D_r$, the real data, for training $C$ cannot lead to gains in performance, because of the additional layer of approximation that $G$ introduced.

Any improvements that one observes when using $D_s$ instead of $D_r$ are not due to systematic principles, but can only reflect artifacts (such as overfitting of $C$ when using $D_r$) or random effects (implying that $C$ may not be able to approximate the distribution when using $D_r$ as well as when using $D_s$). If the generator G is pre-trained on another data set this upper limit does no longer exist.

In summary, it makes sense to evaluate the quality of generated data in terms of how much of the accuracy of the classifiers achieved on real data one can recover when replacing the real data with synthetic data. We perform this evaluation by establishing *recovery rate* $R(a_r, a_s)$ as per the following definition:

$$R(a_r, a_s) = \frac{a_s}{a_r} \tag{2}$$

where $a_r$ is the test accuracy when a classifier is trained on the full real training data and $a_s$ is the test accuracy when the same classifier is trained on a synthetic data set. Based on the above reasoning, one can expect that $R(a_r, a_s) \in [0, 1]$, unless artifacts or random effects disturb the training processes in general.

Note already here that we demonstrate that the generated data do not merely reproduce the real data using other metrics (see just below for the definitions, and Section 4 for the corresponding experiments).

### 3.4.2 NEAREST NEIGHBOUR ADVERSARIAL ACCURACY

It was shown that diffusion models, when provided with too little training data, tend to reproduce training data instead of generating fresh samples (Somepalli et al., 2023). To quantify at what rate we are affected by such effects, we follow the approach presented by Yale et al. (2019).

$$AA_{truth} = \frac{1}{n_{truth}} \sum_{i=1}^{n_{truth}} \mathbf{1}(d_{truth,syn}(i) > d_{truth,truth}(i))$$

$$AA_{syn} = \frac{1}{n_{syn}} \sum_{i=1}^{n_{syn}} \mathbf{1}(d_{syn,truth}(i) > d_{syn,syn}(i)) \tag{3}$$

$$PrivacyLoss = AA_{truth_{tr},syn} - AA_{truth_{te},syn}$$

where $\mathbf{1}$ is the indicator function. Scores of $AA_{truth} = 0.5$ and $AA_{syn} = 0.5$ mean that this metric cannot distinguish $syn$ from $truth$ (and vice versa). Scores closer to 0 reflect over-fitting, whereas scores closer to 1 reflect under-fitting. Using true training and held out test data $truth_{tr}, truth_{te}$ one can compute privacy loss as $AA_{truth_{tr},syn} - AA_{truth_{te},syn}$. We do not report $AA_{truth,syn} = \frac{1}{2}(AA_{truth} + AA_{syn})$ as in the original paper since underfitting on $AA_{truth}$ and overfitting on $AA_{syn}$ can mutually cancel each other, which leads to deceptively good scores despite the poor models that lead to these scores. For further details we refer the interested reader to the original paper Yale et al. (2019).

### 3.4.3 UMAP

Another common way of evaluating quality of generated data is the visualization of the neighbourhood structure using algorithms like UMAP (McInnes et al., 2018) or T-SNE (van der Maaten & Hinton, 2008). We employ these visualization tools in this paper.

### 3.4.4 CLASSIFIER TRAINING

The most important question that we would like to answer is to what degree replacing true training data with synthetically generated training data leads to losses in prediction. For obtaining answers, we consider two scenarios.

First, we focus on predicting the prevalence of the genetic disease Amyotrophic Lateral Sclerosis (ALS), for which the first whole-genome based classifier was presented only recently (Luo et al., 2023). There, training data was selected from 10 405 individual genotypes, referring to 3192 cases, that is individuals affected by ALS, and 7213 controls, that is individuals not affected by ALS, as determined by medical professionals. Further, ALS is known to be a complex and hard to disentangle genetic disease, which means that reliable classification does not depend on a small set of genes. This was documented in (Luo et al., 2023), by showing that good performance could only be established when employing at least on the order of $10^3$ genes. Unlike in (Luo et al., 2023), we train a common multilayer perceptron (MLP) as a binary (ALS or not) classifier. Note that performance rates achieved here exceed the performance rates displayed in (Luo et al., 2023).

Second, we consider the 2504 genotypes provided through the 1000 Genomes (1KG) project (Consortium et al., 2015) (stage 3), and the one of 26 population labels assigned to the genotypes, which gives rise to a classification task referring to one of 26 different labels. Again, we establish our primary classifier as an MLP, which parallels the situation for the ALS data.

To demonstrate that favorable usage of synthetically generated data does not depend on a particular type of classifier, we further experiment with a transformer based ("Transformer") and a convolutional neural network ("CNN") based classifier.

We evaluate each of the classifiers, trained with true data on the one hand, and synthetic data, as generated by the Diffusion Model, on the other hand, on held out test data (ALS: balanced, 520 positive/520 negative genotypes; 1KG: 10% of total data, unbalanced, haplotypes). We follow the experimental protocol presented in (Luo et al., 2023) for the ALS data.

## 4 EXPERIMENTS

In this chapter, we will evaluate the Diffusion Model according to the metrics outlined before.

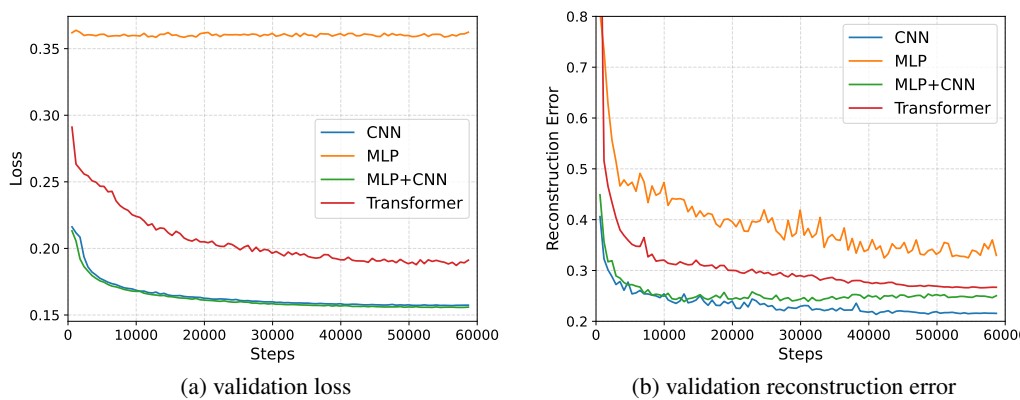

(a) validation loss          (b) validation reconstruction error

Figure 3: We display validation loss (a) and reconstruction error (b) e.g. $||x_p - x||$ during training using single shot denoising.

## 4.1 TRAINING METRICS

First, we show loss curves during training on validation data for the different diffusion model types, see Figure 3. We observe that none of the models over-fit and all of them reduce the reconstruction error $||x - x_p||$ ($x_p$ is the one shot denoising result, see the Appendix for more details) as well as the loss continuously during training.

A closer look at the reconstruction error of the model, visualized in Figure 3 b), shows some interesting results. The MLP model performs only slightly worse compared to the CNN model in reconstruction error, and even though only small drops in loss during training can be observed for the MLP, the reconstruction error keeps improving. The Transformer based architecture seems to be performing well in terms of loss and reconstruction error. We note that reconstruction error is closely related to loss, but scaled by the noise schedule $t$, which means this error focuses more on large values of $t$. For further analysis we refer the interested reader to the Appendix for a discussion of the diffusion process.

## 4.2 EVALUATION

**Disease and population classification**  First, we evaluate different classifiers trained on synthetic data in terms of recovering performance rates that one can achieve on true data—we recall that performance rates achieved on true data establish upper bounds for synthetic data generating mechanisms. See Table 2 for the corresponding results.

We observe that the combined MLP + CNN (U-Net) type architecture outperforms other generator architectures on both 1KG and ALS data, with near-perfect recovery rates on the ALs data. For ALS, the difference between MLP and MLP + CNN generated data is small, but on 1KG data, MLP + CNN clearly outperforms the other architectures. Synthetic data generated by Transformer and CNN based U-Net architectures perform poorly when used for training classifiers. See the Appendix for the accuracy values on which recovery rates are based.

Secondly, we consider the (ubiquitous) scenario where a translational geneticist is forced to restrict her-/himself to a set of available true genotypes that is too small to train reliable classifiers. Here, we evaluate how augmenting small sets of real "seed" training data with larger amounts of synthetically generated data—which may be easily and cheaply available for the particular user—improves performance rates.

See Table 3 for the corresponding results. For example, augmenting only 5% of the real training data (leading to 70% accuracy when used in isolation for training) with synthetically generated data to the overall full amount of data nearly re-establishes the accuracy when using the full, real training data set. This means that the availability of sufficiently large synthetic data sets may rescue efforts

of researchers that remain with too little training data sets due to, for example, budget constraints or restrictive access regulations.

Table 2: Recovery rates on a hold out test set of true genotypes after training different ALS or 1KG population classifiers (MLP, Transformer or CNN) on different synthetically generated data types (generated by: MLP, Transformer, CNN, MLP + CNN). The best synthetic data for each classifier type is marked in **bold**.

|  | Classifier | CNN | MLP | MLP + CNN | Transformer |
|---|---|---|---|---|---|
| ALS data | MLP | 71.51 | 96.69 | **99.49** | 73.77 |
|  | Transformer | 66.06 | 93.44 | **99.41** | 69.30 |
|  | CNN | 69.88 | **91.72** | 91.46 | 68.72 |
| 1KG data | MLP | 15.58 | 65.80 | **93.02** | 13.28 |
|  | Transformer | 16.23 | 62.99 | **84.98** | 8.38 |
|  | CNN | 19.52 | 56.57 | **77.54** | 21.21 |
|  | Average (all) | 43.17 | 78.06 | **90.98** | 42.56 |

Table 3: Accuracy improvements by integration of best synthetic data for best performing classification architecture.

|  | amount of real data | 5% | 10% | 20% | 50% |
|---|---|---|---|---|---|
| ALS Data | no synthetic data | 70.96 | 76.50 | 80.90 | 84.60 |
|  | with synthetic data | 84.83 | 85.01 | 85.34 | 85.70 |
| 1KG data | no synthetic data | 29.01 | 43.99 | 71.52 | 85.19 |
|  | with synthetic data | 83.98 | 86.93 | 87.11 | 87.50 |

**Nearest Neighbour Adversarial Accuracy (NNAA)**   We further evaluate the nearest neighbour adversarial accuracy and Privacy Loss (see Eq. 3), in Table 4. We observe that the combined MLP + CNN (U-Net) architecture performs well in terms of both nearest neighbour adversarial accuracy, and Privacy Loss. Of note, also Transformer and CNN generated data points deliver similar but slightly worse performance in terms of these metrics. We draw two conclusions from this:

Interpreting the experiments, we conclude that the generated data is of sufficiently good quality, in particular that for the combined MLP + CNN architecture, documented by most scores being sufficiently close to 0.5. Improvements are conceivable, however, because the amount of training data used for training generators is likely at the lower limit of quantities required for sound estimation of such high dimensional and complex distributions.

To further quantify the preservation of privacy, we calculated L1, L2, and cosine distances between synthetic and real data points. Thereby, we can confirm that none of the synthetic data points matches any of the real data points. Note that this finding is crucial for maintaining the integrity of diffusion models in terms of privacy (i.e. diversity from a general perspective, which is also reflected in the NNAA score).

In summary, we conclude that the combined MLP + CNN U-Net architecture leads to synthetic genotypes/haplotypes that not only re-establish excellent performance in terms of classification, but also preserve the privacy of the real data used for training the generators to a sufficiently reliable degree.

Table 4: Result of Nearest Neighbour Adversarial Accuracy for generated datasets on the ALS and 1KG data; For AA the closer to 0.5 the better, For Privacy Loss the closer to 0 the better; best performance is **highlighted**.

|  |  |  | CNN | MLP | MLP + CNN | Transformer |
|---|---|---|---|---|---|---|
| ALS data | test data | $AA_{truth}$ | 0.735 | 0.255 | **0.485** | 0.92 |
|  |  | $AA_{syn}$ | 0.68 | 1.0 | 0.93 | **0.66** |
|  | train data | $AA_{truth}$ | 0.81 | 0.005 | **0.405** | 0.93 |
|  |  | $AA_{syn}$ | **0.67** | 1.0 | 0.92 | 0.69 |
|  |  | Privacy Loss | 0.0325 | 0.125 | 0.0475 | **0.02** |
| 1KG data | test data | $AA_{truth}$ | 0.76 | 0.0 | **0.63** | 0.345 |
|  |  | $AA_{syn}$ | 0.995 | 1.0 | 0.94 | **0.92** |
|  | train data | $AA_{truth}$ | 0.765 | 0.0 | **0.385** | 0.285 |
|  |  | $AA_{syn}$ | 1.0 | 0.99 | **0.74** | 0.82 |
|  |  | Privacy Loss | 0.05 | **-0.005** | -0.2225 | 0.08 |

## 5 CONCLUSION

In this work, we have presented, to the best of our knowledge, the first diffusion model based approach by which to generate full-length human genotypes. In this, by standard expansion of genotypes using human genome reference systems, we have also presented an approach by which to generate full-length human genomes at the level of DNA.

In our experiments we have demonstrated, that the synthetically generated genotypes are realistic and that they do not just reproduce the real human genotypes used as input for training.

To demonstrate the practical usefulness of synthetic genotypes, we have employed synthetically generated genotypes as training data for disease- or population-related classifiers. We have shown that such practice re-establishes original performance rates to a degree that justifies their usage in translational genetics research.

Improvements of the diffusion models are readily conceivable by increasing the amount of training data. Note that although limited, one can expect amounts of training data available for training generative models to be larger in real world settings. The reason is that generators can be trained in safe environments, for example as part of the databases that host large numbers of genotype cohorts, which implies that none of the real data has to be shared. Including differential privacy mechanisms may further open up opportunities for usage of generative models in (e.g. federated learning) settings, where sharing parameters of the generative models may be beneficial.

We publish all code at [anonymized for review].

## REPRODUCIBILITY STATEMENT

We publish all code on github and in the supplementary materials used during training and evaluation of the models. The datasets used in the paper were the 1000 Genome Project Consortium et al. (2015) and Project MinE pro (2018). The 1000 Genome data is freely available, while the Project MinE data is only available to credited researchers after a review process.

## ETHICS STATEMENT

Working with human genetic data involves significant privacy and ethical challenges. Our approach aims to mitigate privacy issues by generating synthetic data using diffusion models. However, there remains a potential risk that synthetic samples could inadvertently reveal information about original

data. While we conducted experiments to ensure that our model does not reproduce exact copies of real samples, we cannot fully guarantee that original data cannot be inferred from the synthetic outputs. Future work could incorporate differential privacy mechanisms to provide theoretical privacy guarantees, although this may compromise the model's performance.

We evaluate our diffusion model on real-world classification tasks from previous studies, including the identification of ALS patients, a task that holds promise for future therapeutic advancements. We also use the 1KG Genome ethnicity prediction task, which, while not directly useful to real-world scenarios, serves as a benchmark for evaluating the model's performance on genome-level data.

In general we envisioned the diffusion model being trained in a secure environment and the synthetically generated privatised data being released for further research. To obtain provable privacy, we suggest mechanisms which lead to provable guarantees, such as differential privacy or multiparty computation. As the transfer of SNP data in between different locations might harm privacy, such methods could be combined with federated learning technologies.

A further challenge is a possible bias which can occur due to a skewed training set. Evaluation whether biases exist can be based on downstream tasks. Bias mitigation could be based on resampling strategies for the training data.

Finally, the same as all AI models, we expect that the method is vulnerable to attacks such as data poisoning or adversarial attacks of downstream tasks. Thus, the credibility of data sources and robustness of downstream models needs to be assured. We think, however, that the implementation of these variants is out of the scope of the current paper.

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

## A   APPENDIX

### UMAP VISUALIZATIONS

In Figure 4, data points from real, train and test, as well as synthetically generated data points are visualized using UMAP (McInnes et al., 2018). We observe that almost all generated data points, with the exception of the MLP generated points, are well distributed i.e. hard to differentiate from both test and training data.

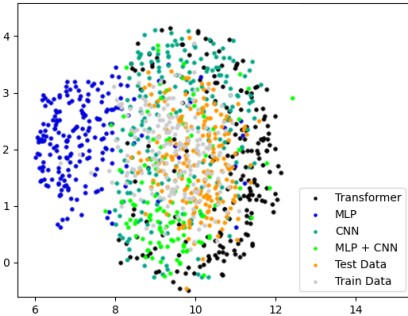

Figure 4: UMAP visualizations of data of different origins. Orange and grey points are test and train data (real). Blue, bright green, black and dark green are MLP, MLP + CNN, Transformer and CNN respectively.

### ANALYZING THE RECONSTRUCTION ERROR OF DIFFERENT MODELS

To further analyze the reconstruction error of different models, we visualize the reconstruction error curves vs amount of noise added of the different diffusion models, in Figure 5. We observe that while the CNN has an overall better performance it focuses more on the fine detail of the structure e.g. recovering from small noise values, while the MLP architecture is more focused on recovering rough structure (e.g. high noise). Furthermore the MLP model does observe almost no improvement during training in recovering fine-details, but solely in recovering rough structure.

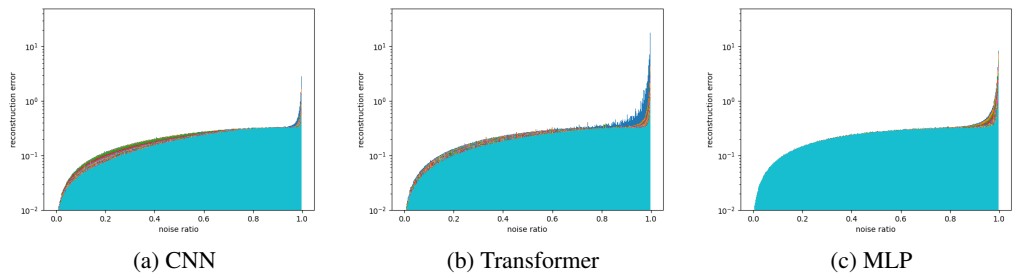

(a) CNN        (b) Transformer        (c) MLP

Figure 5: Reconstruction error vs noise curves during training for different diffusion model backbones. Note the logarithmic scaling of the y-axis. The different colors indicate different epochs during training. The final result has the color light blue. This highlights the improvements during training that the model experiences.

We also see this confirmed when looking at the $\lambda$ parameter found for the MLP+CNN network version, see Figure 6 (a). We observe in Figure 6 (b) how the two parts of the network complement each other especially after longer training times.

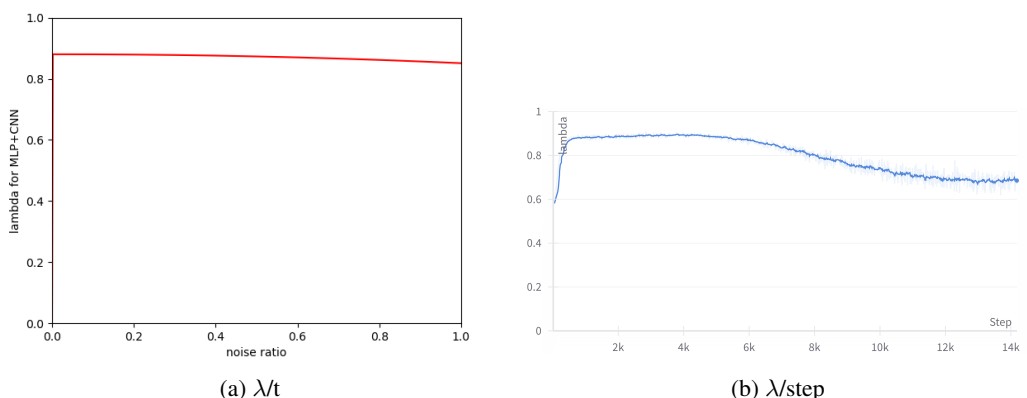

(a) $\lambda$/t             (b) $\lambda$/step

Figure 6: On the left, $\lambda$ vs t (noise amplitude) curves after training for the MLP + CNN diffusion model backbones. Higher lambda results in more weight on the CNN part of the network. On the right, $\lambda$ averaged over noise levels on y-axis and training step on x-axis during training for the MLP + CNN diffusion model backbones. Higher lambda results in more weight on the CNN part of the network.

In Figure 7 we analyze various bottlenecks of our diffusion models. The MLP based diffusion model as depicted in Figure 7 a) has trouble finding much structure in the data and only clearly differentiates MLP + CNN generated data from the rest.

The CNN based diffusion model as depicted in Figure 7 b) finds more structure and difference between data of synthetic origin and real data.

Both models on their own are not very accurate at separating the Transformer based data or their own generated data from the original data. We postulate that this might be a reason why they complement each other well in the hybrid MLP + CNN architecture.

However, it is unclear why both models have trouble separating the Transformer based data from the train and test data. This would generally indicate that the Transformer based data is of high quality, which other metrics (NNAA, Recovery Rates) in this paper disagree with.

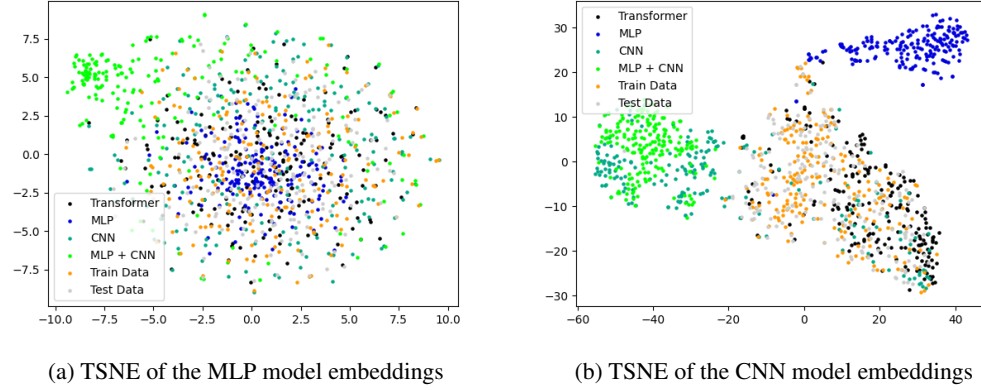

(a) TSNE of the MLP model embeddings       (b) TSNE of the CNN model embeddings

Figure 7: On the left, a TSNE dimension reduction of the bottleneck of the MLP diffusion model of 200 genomes from various data sources. The same on the right for the CNN diffusion model.

ARCHITECTURE DETAILS

In this section we present some technical details in more depth than possible in the main paper.

Table 5: Technical details for all generative models. Tflops ($10^{12}$ flops) were determined using Pytorchs profiler. Training time was measured on a single Quadro 5000 RTX in seconds for the ALS data set. Units are given in brackets ().

|  | CNN | MLP | MLP + CNN | Transformer |
| --- | --- | --- | --- | --- |
| Training time (s) | 45.000 | 8.000 | 52.000 | 58.000 |
| Parameter count | 18.5561.68 | 310.094.848 | 328.651.401 | 135.280.130 |
| TFlops per genome (Tflops) | 1.73 | 0.62 | 2.35 | 49.29 |

*CNN*

The CNN diffusion model we use is very similar to the one used in stable diffusion (Rombach et al., 2022) and looks structurally similar to the MLP Unet. It is made up of 1D Convolutions instead of 2D, has 8 downsampling blocks with channel multipliers (1,1,1,1,2,2,3,4) and multi head attention at blocks 5 and 6. The base filter size used in our CNN experiments is 64. See Figure 8 for more details. A very deep convolutional architecture was used with a relatively large number of down and up blocks to account for the large sequence length of  18.000 of the genome data compared to typical image sizes of  1.000. In particular we want to point out that we experimented with a linear layer for each PCA in front of the CNN architecture. This was done to project the Genomes PCA into a shared embedding-space. Sadly this did not achieve superior performance.

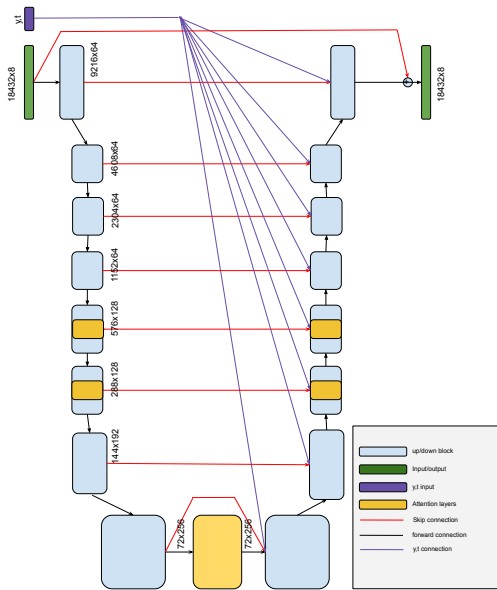

Figure 8: An overview of the CNN diffusion model architecture.

*MLP*

See Figure 2 in the main paper, for a good overview of the MLP diffusion model.

*Transformer*

The transformer architecture we use is very similar to the one used in ViT (Dosovitskiy et al., 2020), we use 12 encoder layers with feature size of 384, see Figure 9 for more details. The conditioning in the form of $t$ and $y$ is injected using 2 additional tokens. The first and last linear projection layers don't have shared weights for every position as in ViT, but rather are unique to every position. This was done as in contrast to the image domain where pixels always encode the same information, SNPs do encode different genetic mutations depending on position.

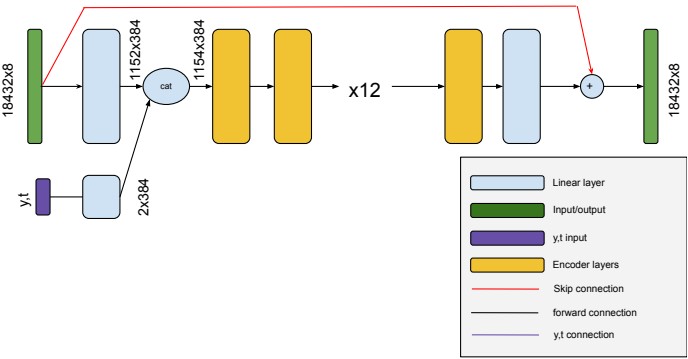

Figure 9: An overview of the transformer diffusion model architecture.

ADDITIONAL RESULTS

Here we display the results from section 4 not as recovery rates but as accuracy's.

Table 6: Accuracy on a hold out test set after training different ALS or 1KG population classifiers (MLP, Transformer or CNN) on different synthetically generated data types (generated by: MLP, Transformer, CNN, MLP + CNN or Baseline). The best synthetic data for each classifier type is marked in **bold**.

| Classifier | Real Data | CNN | MLP | MLP + CNN | Transformer |
|---|---|---|---|---|---|
| | | | ALS data | | |
| MLP | 87.60 | 62.64 | 84.71 | **87.15** | 64.62 |
| Transformer | 82.11 | 54.23 | 76.73 | **81.63** | 56.92 |
| CNN | 73.17 | 51.15 | **67.11** | 66.92 | 50.29 |
| | | | 1KG data | | |
| MLP | 90.23 | 14.06 | 59.38 | **83.98** | 11.98 |
| Transformer | 74.61 | 12.11 | 47.01 | **63.41** | 6.25 |
| CNN | 77.34 | 15.10 | 43.75 | **59.99** | 16.41 |
| Average (all) | 80.84 | 34.89 | 63.12 | **72.94** | 34.41 |

DISCUSSION ON HUMAN GENOTYPES.

The human genome is a sequence of approximately 3 billion letters $\{A, C, G, T\}$, reflecting the nucleotides that form the basis of DNA. The vast majority of the 3 billion letters in human genomes are identical across all individual genomes; only approximately 3-5 million positions, referred to as *polymorphic sites*, vary. Although all types of variations can be observed at such polymorphic sites, the vast majority of such sites exhibit single nucleotide polymorphisms (SNPs), defined by exchanges of single letters. In the following, we will only deal with SNP sites, and neglect polymorphic sites characterized by other types of mutations. Restricting oneself to studying SNPs is well justified, because the vast majority of other sites are in *linkage disequilibrium (LD)* with SNP sites. That is, in other words, they are statistically strongly correlated with SNP patterns nearby, such that one can safely infer non-SNP site contents based on knowing the SNPs of an individual.

Further, the great majority of SNP sites are *biallelic*, which means that only two letters happen to appear in human genomes; for example, a particular site may be defined by an $A$ showing in some genomes, and a $T$ showing in the other genomes. The letter that appears in the

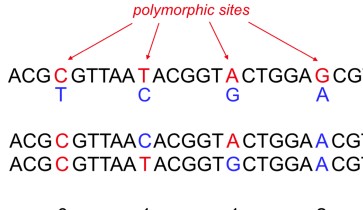

Figure 10: The (unphased) genotype counts the number of alternative alleles (blue) in the two ancestral genome copies each individual inherits.

majority of people is referred to as *reference allele*, whereas the letter showing in the other, minor fraction of people is referred to as *alternative allele*. Every human genome consists of two copies, of which one is inherited from the mother, and the other from the father. Of course, the contents of the two copies can vary at the SNP sites: the reference allele can show in both copies, referred to as *homozygous for the reference allele*, the reference allele can show in one copy while the alternative allele shows in the other copy, referred to as *heterozygous* (for example, while the mother genome copy exhibits the reference allele $G$, the father copy exhibits the alternative allele $C$, or vice versa) or the alternative allele can show in both copies, referred to as *homozygoous for the alternative allele*.

Let $N$ be the number of SNP sites in human genomes. The *genotype profile $G$* is a vector of length $N$ over the entries $\{0, 1, 2\}$, where $0, 1, 2$ refer to alternative allele counts at the SNP sites. That is, $0$ reflects a polymorphic site that is homozygous for the alternative allele, $1$ refers to a heterozygous site, and so on. For example, let $i \in \{1, ..., N\}$ refer to the $i$-th SNP site, and let $G$ be the genotype of an individual, then $G[i] = 1$ reflects that the individual that gives rise to $G$ inherited the reference allele from one of the ancestors, while having inherited the alternative allele from the other ancestor.

Note that expanding the genotype $G$ for an individual into a full-length genome over the alphabet $\{A, C, G, T\}$ corresponds to a straightforward operation: for any non-SNP polymorphic site, insert the by LD principles statistically most likely variant, and for any non-polymorphic site, insert the only applicable reference letter. While not necessarily matching the real genome that gave rise to $G$ in all places, the genome resulting from this expansion operation is highly likely to reflect true genetic sequence in the great majority of places.

In our work, the input to our diffusion model are individual genotype profiles $G \in \{0, 1, 2\}^N$, where $N$ is the number of SNP sites. Individual genotype profiles are the by far predominant way of representing human genomes in the frame of genome wide association studies (GWAS). In the meantime, large databases have been filled up with individual genotypes of this kind. Of course, access to such genotype profiles is subject to stringent access regulations, because it is straightforward to match genotype profiles with individuals in a unique manner—in fact, already the content of 50 (sufficiently distant) SNP sites suffices to uniquely identify single individuals.

Note that untreated genotype profiles are still too large to serve as input to diffusion models. Preferably, one can trim down the length of the input from several (3-5) millions to only several tens of thousands, that is by a factor of 100. Below, we describe how to embed genotypes into real-valued vector spaces of dimension in the tens of thousands.

## DIFFUSION MODELS

In this section we provide a brief overview of the diffusion process that we have implemented. For details on diffusion processes in general, please see Ho et al. (2020); Song et al. (2022).

## TRAINING

Although originally presented in Sohl-Dickstein et al. (2015), diffusion models have only recently gained popularity. As generative models, they reflect frameworks that learn probability distributions from data that are too complex to draw samples from in other ways. The training procedure is driven by iteratively adding Gaussian noise to known examples and, simultaneously, learn the way back by means of a neural network (NN). After training, sampling reflects to sample pure noise, and predict the way back to the real data distribution by means of the trained NN. Formally, one iteration adds Gaussian noise $\epsilon = N(\mu, \sigma)$ with variance $\sigma$ and mean $\mu = 0$ to real data $x \in D$ sampled from Distribution $D$, thereby generating a sequence of ever noisier $x_t, t \in (0, T]$, referred to as the *forward process*. Eventually, $x_T$ can no longer be distinguished from pure Gaussian noise; in parallel, the NN seeks to learn how to return from $x_t$ to $x_{t-1}$ for $t \in (0, T]$, which is referred to as the *reverse process*.

We introduce the auxiliary variables

$$\alpha(t) = 1 - \beta(t); \ \overline{\alpha}(t) = \prod_{s=1}^{t} \alpha_s; \ \tilde{\beta}(t) = \frac{1 - \overline{\alpha}(t-1)}{1 - \overline{\alpha}(t)} \beta(t) \tag{4}$$

where here $\beta_t$ corresponds to the variance that refers to the noise added when moving from $x_{t-1}$ to $x_t$. In practice (see Ho et al. (2020)), the $\beta_t$ are pre-determined and follow a linear schedule, increasing from $\beta(1) = 10^{-4}$ to $\beta(T) = 0.02$. Steps $x_{t-1}$ to $x_t$ can be summarized into one formula, yielding

$$x_n(t) = \sqrt{\overline{\alpha}(t)} \cdot \epsilon + \sqrt{1 - \overline{\alpha}(t)} \cdot x \tag{5}$$

The process is designed for a data distribution $D$ with mean $\sigma = 1$ and variance $\mu = 0$, which can easily be achieved by pre-processing steps.

In practice, the NN, $E$ is only required to predict the noise $\epsilon_t$ that was added in each step $t \in (0, T]$; we refer to the predicted noise as $\epsilon_p$. Formally, $E$, upon having received timestep $t$ and the noisy example $x(t)$ as input, predicts

$$\epsilon_p = E(t, x(t)) \tag{6}$$

which when subtracting $\epsilon_p$ from $x(t)$ yields a denoised version of $x(t)$. Recovering the the original data $x$ in a single step can be computed according to:

$$x_p(t, x) = \frac{x_n(t) - (1 - \sqrt{\overline{\alpha}(t)}) \cdot \epsilon_p}{\sqrt{\overline{\alpha}(t)}} \tag{7}$$

The loss function $L(x)$ of our neural network reflects to correctly predict the added noise:

$$L(x) = ||\epsilon - \epsilon_p(x)|| \tag{8}$$

where, in our case, $|| \cdot ||$ reflects the L2 norm, which is a common choice. Sampling $t$ during training follows a uniform distribution over drawing $t \in \{1, ..., T\}$

DATA GENERATION

Generating new data is done analogously to recovering the original data. Starting from complete noise $x_{n,T} = N(0, I)$, the NN iteratively predicts the noise to be removed, which leads to a data point $x_n = x_{n,0}$ that stems from the original distribution:

$$x_{n,t-1} = \frac{1}{\sqrt{\alpha(t)}} \cdot (x_{n,t} - \frac{1 - \alpha(t)}{\sqrt{1 - \overline{\alpha}(t)}} \epsilon_p(x_{n,t}, t)) + \sqrt{\tilde{\beta}(t)} N(0, I) \tag{9}$$

This general process, as described in detail in (Ho et al., 2020), has been repeatedly pointed out as being successful in generating realistic artificial images, for example. Applying this process to generating genotype profiles (or their embeddings), which one can easily expand into full-length genomes, establishes a novelty.

In addition to this basic process, one can further incorporate conditioning information $y$ by changing Eq. equation 6 to

$$\epsilon_p = E(t, x_n(t), y) \tag{10}$$

Typically $y$ reflects an input vector that specifies additional information about the data sample $x$. For example, in the case of an image $x$, this could be the caption of the image, or whether or not the image contains particularly labeled elements, such as, for example, trees or beaches as part of the

image. Providing $y$ along with $x$ drives the generation process towards the generation of new data that takes the additional information into account, so, when following our example further, generates images that contain trees or beaches. In our work, $y$ refers to labels that characterize human genomes in terms of population or disease phenotypes.

