# OpenReview forum: "Generating Synthetic Genotypes using Diffusion Models"
_ICLR.cc/2025/Conference — Submitted to ICLR 2025_

### Official Review · Reviewer_ydeS · 2024-11-01

**Soundness:** 2
**Presentation:** 1
**Contribution:** 2
**Rating:** 5
**Confidence:** 3

**Summary:**

In this paper, the authors use diffusion models to synthesize genomic data, allowing researchers to access it at a low cost. They expanded the synthesis scale to encompass the entire human genome. To handle long sequences, the authors employed a method similar to Latent-Space Diffusion Model: first splitting the genome into parts. For each part, they used PCA as an encoder/decoder to reduce the dimensionality to 8. The authors applied various methods to test the model's generation accuracy, generalization performance, and efficiency in downstream tasks. Notably, models trained on synthetic data performed almost identically to those trained on real data.

**Strengths:**

As the authors claim, they propose the first model that augments human genomic data at full scale, which is larger compared to previous works. Their experiments demonstrate that their generated data can be used as effectively as real data to train downstream models.

**Weaknesses:**

The paper's most significant weakness is its clarity and writing quality. In the section on previous works, there is only a table without main text, making it difficult to understand how the authors' approach compares to prior research. What are the limitations of previous works beyond their smaller scale? How does the authors' approach relate to existing methods?

The authors' claim of "using diffusion model" as a novelty is questionable. Diffusion models aren't fundamentally different from other generative models. While using a diffusion model might be justified based on performance, it doesn't constitute a novel contribution in itself.

Lines 92 to 94 are confusing: they state that DDIM is beneficial, but DDPM is also necessary, without propose any solutions. In fact, DDPM is a special case of DDIM with specific alpha and sigma values.

On line 95, the authors mention using classifier-free guidance, claiming it can improve quality. However, classifier-free guidance was developed for conditional generation, not specifically for quality improvement. The authors should clarify their reasoning for this claim.

The experiment section lacks details about the data generation process. Although the model was trained with classifier-free guidance, it's unclear how the data is generated with conditions. The authors should clarify how they generate the data and how they select the conditions for generation.

Several other details also require improvement. The UMAP experiment is distracting and should be removed, as it's highly subjective and doesn't provide useful information. In Figure 2, the horizontal lines are not truly horizontal, and the markers are misplaced. The notations in this figure are unclear, making it difficult to determine which parts they refer to. Figure 3 appears to be a screenshot from Weights & Biases, which is unprofessional. Figure 5 is challenging to read due to its poor layout and formatting.

Regarding the model design, the use of CNNs is questionable due to the lack of translational symmetry. The same vector can represent entirely different information in the 18,279 or 26,624 distinct latent spaces created by PCA.

**Questions:**

My biggest question concerns how exactly you augment the data. There are no details about how you utilize the conditional generating capabilities. Given a small amount of data, how do you select the conditions? What is the relationship between your "seed" training data and the generated data?

---

> ### Author Response · Authors · 2024-11-19
>
> We sincerely appreciate your overall judgment of our work. At the same time, we regret that so far we have not been able to convey the technical details that you have pointed out. In the revised version, we will address all your concerns in sufficient detail.
>
> Please find a point-by-point response to all Weaknesses you pointed out and all Questions you asked.
>
> Questions:
>
> Data augmentation: In fact, we do not augment our data. Data augmentation methods are not as widely spread as, for example, in the vision domain. Because they can introduce uncertainties, we refrain from augmenting data. We remain somewhat worried: why do you assume that we augmented our data in any way? Please let us know so that we can implement the necessary improvements.
>
> Conditioned data: In our case, conditioned data refers to non-diseased (control) genomes on the one hand, and diseased (case) genomes on the other hand. We apologize for not having pointed out sufficiently detailed how conditioned data is generated. In fact, generating conditioned data follows the protocol described in the classical works on diffusion models [1,2], and refers to the integration of labels as displayed via the ‘y’ variable in Figure 2.
>
> Selecting conditioning: We select this conditioning to be uniformly random distributed. That is, ALS positive and ALS negative are equally likely to be selected for each generated sample. For the 1KG dataset we follow the same principle, that is, each of the 26 classes is equally likely to be chosen for generation. We have added a description of these details to the appendix.
>
> Seed training data: The seed training data is a randomly selected subset of the true data, too small to warrant successful training of classifiers in its own right. Augmenting (small amounts of) true seed data with (substantially larger amounts of) artificially generated data for training leads to substantial performance improvement, see Table 3.
>
>
> Weaknesses:
>
> Discussion of related work: Thanks for pointing this out. In the revised version, we have added a more detailed discussion of related work to the manuscript.
>
> Novelty: We agree that diffusion models are not a novelty per se, and we apologize for not having this conveyed appropriately.
> What we really would like to convey is the following: using diffusion models for generating whole human genomes, as well as studying possible novel architectures needed for such models does constitute a novelty that is (highly) relevant in the area of genomics / biomedical applications. Note that all other generative models for generating genome data have failed to generate full-length genome data. In the revised version, we try to highlight this more decidedly.
>
> DDIM: We are using DDPM thanks to the higher quality that one achieves when using full step DDPM generation.  We agree that DDIM has benefits with respect to speed, which, however, we found negligible in our case: these benefits do not offset the benefits in terms of quality when using DDPM. Please have a look at the original DDIM paper [1]: they explicitly write that DDIM trades speed for quality in comparison with DDPMs (see the Abstract). We have revised the manuscript accordingly to clarify this.
>
> Classifier free guidance: This is a technique to guide the generation process towards more archetypical samples of a class i.e. establishes a tradeoff between coverage of modes and fidelity of samples (see [2]). We agree that classifier-free guidance addresses conditional generation, and not quality improvements. We wanted to allude to improvements in terms of conditional generation in comparison with the standard conditional generation procedure. In [3] classifier-free guidance is used to increase the performance of classifiers trained on synthetically generated data, subsequently evaluated on real data.
>
> Details: We have taken all weaknesses that you pointed out into thorough consideration, and have revised the manuscript accordingly. We would like to thank for pointing them out.
>
> Translational symmetry: We address this lack of translational symmetry in lines (219-229) and point out that this is a weakness of this kind of architecture. The datasets have a consistent dimensionality. This enables these kinds of architectures to still work, and even outperform transformer architectures on the datasets at hand here.
>
> [1] Jiaming Song, Chenlin Meng, and Stefano Ermon. Denoising diffusion implicit models, 2022.
> [2] Jonathan Ho and Tim Salimans. Classifier-free diffusion guidance. arXiv preprint arXiv:2207.12598, 2022.
> [3]Shekoofeh Azizi, Simon Kornblith, Chitwan Saharia, Mohammad Norouzi, and David J. Fleet. Syn-thetic data from diffusion models improves imagenet classification. Transactions on Machine
> Learning Research, 2023.

---

> > ### Comment · Reviewer_ydeS · 2024-11-22
> >
> > > Novelty: We agree that diffusion models are not a novelty per se, and we apologize for not having this conveyed appropriately. What we really would like to convey is the following: using diffusion models for generating whole human genomes, as well as studying possible novel architectures needed for such models does constitute a novelty that is (highly) relevant in the area of genomics / biomedical applications. Note that all other generative models for generating genome data have failed to generate full-length genome data. In the revised version, we try to highlight this more decidedly.
> >
> > I critise is is mainly because I don't think "method-driven" can lead to any novelty. A proper way should focus more on questions or hypothesis, rather than "we use B on X, and previous people didn't do it".
> >
> > > DDIM: We are using DDPM thanks to the higher quality that one achieves when using full step DDPM generation. We agree that DDIM has benefits with respect to speed, which, however, we found negligible in our case: these benefits do not offset the benefits in terms of quality when using DDPM. Please have a look at the original DDIM paper [1]: they explicitly write that DDIM trades speed for quality in comparison with DDPMs (see the Abstract). We have revised the manuscript accordingly to clarify this.
> >
> > I don’t think you have fully understood the relationship between DDIM and DDPM. The reason I mentioned, ‘In fact, DDPM is a special case of DDIM with specific alpha and sigma values,’ is because you can select specific $\alpha$ and $\sigma$ values to make DDIM exactly equivalent to DDPM. Refer to the DDIM paper, page 5, below equation (12), where they explicitly state this.
> >
> > "When $\sigma_t=\sqrt{(1-\alpha_{t-1})/(1-\alpha_t)}\sqrt{1-\alpha_t/\alpha_{t-1}}$ for all $t$, the forward process becomes Markovian, and the generative process becomes a DDPM."
> >
> > And this is why I don't think you should compare them in a opposite way.
> >
> > > Classifier free guidance: This is a technique to guide the generation process towards more archetypical samples of a class i.e. establishes a tradeoff between coverage of modes and fidelity of samples (see [2]). We agree that classifier-free guidance addresses conditional generation, and not quality improvements. We wanted to allude to improvements in terms of conditional generation in comparison with the standard conditional generation procedure. In [3] classifier-free guidance is used to increase the performance of classifiers trained on synthetically generated data, subsequently evaluated on real data.
> >
> > > Details: We have taken all weaknesses that you pointed out into thorough consideration, and have revised the manuscript accordingly. We would like to thank for pointing them out.
> >
> > Thank you for clarifying this and update the paper.
> >
> > >  Translational symmetry: We address this lack of translational symmetry in lines (219-229) and point out that this is a weakness of this kind of architecture. The datasets have a consistent dimensionality. This enables these kinds of architectures to still work, and even outperform transformer architectures on the datasets at hand here.
> >
> > For symmetry, I don’t think your update resolves the issue. PCA or SVD do not inherently make the dimensions of the dimension-reduced vector comparable. A CNN trained for PCA 1 will not necessarily perform well on PCA 2 (as shown in Figure 1). A more natural approach might involve adding learnable linear transformations across all PCA dimensions. Without this, the effectiveness of the CNN remains questionable.
> >
> > Lines 226–231 introduce additional issues. You assert that transformers lack spatial biases, which is a puzzling claim. Spatial encoding was addressed in the original transformer paper through the use of positional encoding. Consequently, the comparison with transformers appears to be unfair.
> >
> >
> >
> > I have raised your score and will further increase it to a positive level if you can address these points effectively.

---

> > > ### Author Response · Authors · 2024-11-24
> > >
> > > DDIM vs DDPM:
> > > Thank you for pointing this out. We agree with your clarification regarding the relationship between DDIM and DDPM, and we will revise the paper to ensure we are not presenting them in an oppositional manner. Our intent was to explain exactly how the generation of the synthethic data works, particularly to preempt questions such as, “Why not use DDIM?” We value this feedback and will refine the text to better reflect the equivalence between DDPM and DDIM under specific parameter choices, as outlined in the DDIM paper.
> > >
> > > Symmetry:
> > > We appreciate your insights regarding the embedding space and comparability of PCA dimensions. During our research, we did experiment with the exact approach you suggested—a learnable linear transformation across all PCA dimensions before the CNN. We agree that, in principle, this should enhance performance by embedding the PCA dimensions into a shared space.
> > > However, in practice, this architectural modification hindered the training of the CNN, leading to suboptimal performance. While we did not include these details in the main text due to space constraints, we will add a discussion in the appendix to elaborate on this observation. Exploring why this happens remains an interesting avenue for future work.
> > >
> > >
> > > Transformer spatial bias:
> > > We apologize for any confusion caused by our original phrasing. Our intention was to highlight that Transformers do not inherently encode spatial bias. However, as you correctly noted, positional encodings (including learned ones, as used in our study) reintroduce this spatial information. We will revise the manuscript to clarify this distinction.
> > >
> > > With regards to fairness of comparison:
> > > Regarding the fairness of comparisons between architectures, we fully acknowledge that Transformers leverage positional encodings to encode spatial information.
> > > The CNN-only architecture, on the other hand, lacks this inherent advantage, which likely contributes to its poorer performance in classification tasks.
> > > Our MLP+CNN architecture, however, incorporates spatial information via the MLP, which allows the model to focus on specific genomic positions.
> > >  Consequently, we believe it is fair to compare the MLP+CNN architecture against the Transformer, as both have mechanisms to encode positional information. The CNN-only architecture, by contrast, serves as a baseline and its lower performance aligns with expectations. We will clarify these points in the manuscript to ensure transparency and fairness in the comparisons.
> > > We greatly appreciate your detailed feedback, which has helped us improve both the clarity and rigor of our work. Your suggestions will be fully incorporated into the revised manuscript.

---

### Official Review · Reviewer_iupA · 2024-11-03

**Soundness:** 1
**Presentation:** 1
**Contribution:** 1
**Rating:** 1
**Confidence:** 5

**Summary:**

In this work, the authors propose a technique for generating complete human genomes. Motivated by the challenges of obtaining individual genomes due to sequencing costs and data privacy concerns, they argue that generating synthetic genomes will improve the performance of predictive models. To address the long-context problem faced by machine learning models processing full genomes, they employ a previously introduced technique to embed genomes in a low-dimensional latent space. A diffusion model is then trained within this space. The authors train their model using classifier guidance on two datasets: one for generating genomes with or without ALS, and another for generating genomes representing 26 populations from the 1000 Genomes Project. To validate their diffusion model, they try to demonstrate (1) that classifiers trained on the generated data achieve comparable accuracy to those trained on real data, and (2) that the generated data distribution is both diverse and sufficiently distinct from the real data distribution.

**Strengths:**

Most current deep learning models in genomics operate at the sequence level, preventing them from processing complete genomes due to their immense length. Additionally, most models rely on the reference genome, hindering their ability to accurately capture intra-individual variants. These limitations, well-documented in the literature, are likely central to the challenges faced in improving these models. Therefore, I appreciate this paper's focus on working at the complete genome level to represent full individuals. Employing an embedding technique to represent full genomes in a low-dimensional space and utilizing latent diffusion within this space is a promising approach, and I am happy to see research in this area.

Furthermore, the choice of datasets for training the model is well-justified.

**Weaknesses:**

However, the paper presents several limitations:

- The most compelling aspect, in my view, is the embedding of complete genomes into a low-dimensional space and subsequent model training. However, the employed technique is not a novel contribution and lacks sufficient discussion. The authors also fail to adequately address the extent to which complete genomes can be reconstructed from this latent space and the impact on the final sequence. While reconstruction is briefly mentioned, a detailed explanation of the process and precise reconstruction metrics at the sequence level are missing.

- The literature review is incomplete. The authors should include references to D3 [1] and DNADiffusion [2]. While these works have slightly different motivations and are not directly comparable, they warrant mention. Given the discussion of long-range limitations in current models, referencing Enformer [3] and Borzoi [4] would also be beneficial as they represent significant advancements in addressing this issue.

- The paper lacks crucial technical details, hindering a comprehensive understanding of the proposed method. Certain sections require rewriting for improved clarity. Specifically, a dedicated section should detail genome processing and embedding, and another should explain the diffusion model training process, including duration. The model architecture section is difficult to follow, and the ablations lack proper introductions. The section introducing the classifier accuracy recovery metric is overly long and imprecise; it could be condensed into a few concise sentences. Section 3.4.2, while critical, is poorly detailed. The introduced metrics lack explanations, particularly the notation 'd' (presumably representing distance), which is not defined, nor is its measurement space specified. Additionally, the mathematical notations in the equations are incorrect. Measuring the quality and diversity of generated data is crucial in generative modeling and has been a key research area in computer vision. I strongly encourage the authors to refine this aspect, provide thorough introductions to their metrics, and better reference existing literature.

- Employing a convolution-based architecture might be suitable, but the authors should reference existing architectures, use them as a starting point, and discuss necessary modifications for their specific context. Inspiration can be drawn from architectures used in genomics, such as those in D3, Borzoi, BPNet, and Enformer. The 1D U-NET architecture from SegmentNT [5] could also be relevant. Additionally, a vast body of literature exists on 1D convolution and 1D U-NET in signal processing.

- The figures are of poor quality. The cartoons are unclear and lack information, and the font sizes are too small. The evolution plots are difficult to read, and using screenshots from run monitoring software is unacceptable. The authors should regenerate these plots using a plotting library like matplotlib.

- The paper's structure is unconventional. The authors underutilize the allotted space and employ numerous small sub-paragraphs. A denser presentation is expected for a conference like ICLR. The authors should utilize the available space to provide a detailed description of the method and experimental setup.

- The experimental section is also unconvincing. The experimental setup lacks a proper introduction; for instance, the classifier's input and architecture are not specified. As previously mentioned, the privacy metrics are inadequately introduced, making it difficult to assess the reported performance. Lastly, the experiment investigating the use of synthetic data for improving predictors in data scarcity regimes is flawed. When training the classifier on k% of the dataset, the diffusion model should also be trained on k% to accurately reflect the intended use case. In reality, with only k% of the data available, training a diffusion model on 100% is not feasible.

Overall, this paper falls short of the expected quality and contribution standards for ICLR, and I confidently recommend strong rejection.

[1] https://www.biorxiv.org/content/10.1101/2024.05.23.595630v1.abstract

[2] https://www.biorxiv.org/content/10.1101/2024.02.01.578352v1

[3] https://www.nature.com/articles/s41592-021-01252-x

[4] https://www.biorxiv.org/content/10.1101/2023.08.30.555582v1

[5] https://www.biorxiv.org/content/10.1101/2024.03.14.584712v1.full.pdf

**Questions:**

See my points above.

---

> ### Author Response · Authors · 2024-11-20
>
> We would like to thank you for your input. While we appreciate your critique, we tend to be somewhat concerned by the overly confident judgment. To the best of our understanding, some of your critics are due to either not having read the manuscript at its full range, or a lack of background knowledge on your end. While we agree with some of the presentational weaknesses you point out, we tend to disagree with you on the majority of the other points. Of course, insufficient writing and presentation may hamper understanding, but misjudging the overall contents at their core feels like an expression of a different kind of issue (given that the other reviewers did not have such misunderstandings). We sincerely ask for revisiting your ideas about our manuscript, expressing our gratitude if you did.
>
> Following, please see a point-by-point response to your comments.
>
> Novelty: Raising a diffusion model to generate full-length genotype data is a novelty from two aspects. First, nobody before has been able to present a generative technique by which to generate full-length human genotypes. Secondly, nobody has used diffusion models on genotype data before, to the best of our knowledge.
>
> Reconstructing genomes from latent embeddings: This follows a two-step procedure. First, genotypes are reconstructed from their PCA representation, which, to the best of our judgment, does not require any particular explanation. Second, reconstructing genomes from genotypes reflects standard genomics practice: nucleotides for non-polymorphic sites can be read off applicable reference genomes, so as to be filled into the genotype skeleton to turn it into a DNA sequence.
>
> Note further that we mention this in passing, just because this is theoretically possible. Here, we decidedly focus on genotypes. The amount of full-length human genotypes raised to date greatly exceeds the amount of DNA level sequence available, in addition to having a clear biomedical purpose, namely to support genome-wide association studies. Genotype data is the standard outcome of any genome-wide association study, which explains its plentifulness.
>
> Putatively incomplete literature review: We generally disagree on including any (but one) of the works mentioned, because they only relate to our work insofar as generating nucleotide sequences matches the idea of generating synthetic genome-related data in the widest sense. We agree on possibly adding DNADiffusion, because diffusion models were used there. In the general interest of inflating reference sections, we would like to refrain from adding an excessive amount of references. All of the citations we have already included are substantially closer related, and adding more would also dilute the targeted message of the paper: generating synthetic genotype cohorts using diffusion models.
>
> Convolution based U-Net: We were well aware of the sequence U-Nets you mention. However, the simple reason for our choices is that employing CNN based U-Nets yielded optimal performance rates.
>
> “Unconvincing” experimental section: The experiments are designed to reflect GWAS practice. We disagree with you reckoning that the amount of true seed genotypes should match the amount of synthetic genotypes (not understanding in turn the logic inherent to this suggestion). We believe, however, although we clearly tried, that we may have not been 100% clear about the real-world scenario we have in mind. There are large databases containing vast amounts of genotype data (raised in the context of GWAS), referring to a plethora of diseases and non-disease phenotypes; the most prominent example is dbGaP [1].
>
> The scenario we envision is as follows: a researcher was able to raise, say, a few hundred genotypes—note that genotype data is quite expensive. Out of budget, the researcher now is left with expensive data that cannot support any GWAS type analysis. This researcher could now request synthetic data from dbGaP, for example: this database can afford training a diffusion model on tens / hundreds of thousands (even millions) of genotypes. While not being allowed to share the original data, this database can then, upon request, provide researchers who remained with too small amounts of local data with synthetic genotypes. Therefore, the database needs to make sure that the synthetic genotypes are both sufficiently realistic and sufficiently different from true genotypes that were used to training the diffusion model.
>
> This scenario is ubiquitous, worldwide: individual researchers remaining with local datasets too small to support statistically justified statements about the diseases they investigate.
>
> Although we already tried to explain this, we will make an effort to improve the illustration of this real-world scenario.
>
> Any other weakness you mention apparently refer to rather minor presentation issues. We will be happy to take care of them.
>
> [1] https://www.ncbi.nlm.nih.gov/gap/

---

> ### Comment · Reviewer_iupA · 2024-11-20
> **Answer to authors comment**
>
> Thank you for your detailed response to my review. I appreciate you taking the time to address my concerns.  While I acknowledge your points, I still have some reservations about the clarity and technical strength of the paper.
>
> I confirm that I read the manuscript carefully, and I had to re-read it several times to understand the core concepts due to the presentation. Regarding background knowledge, I am an active researcher in this field. While self-assessment can be difficult, my colleagues share my concerns about the technical quality and presentation, which we believe fall short of the standards expected at ICLR.
>
> Point-by-Point Response
>
> **Novelty**: I understand your claim about the novelty of generating full-length genotypes using a diffusion model. However, the experiments conducted in latent space make it difficult to assess the feasibility of generating complete and valid genotypes. To convincingly demonstrate this novelty, I recommend presenting validation results on the reconstructed genotypes and full genomes.
>
> **Reconstructing Genomes**:  While reconstructing genotypes from the latent space and subsequently generating full genomes might be theoretically possible, the paper needs stronger empirical evidence to support this claim.  As I suggested in my review, consider adopting a rigorous validation pipeline similar to the one used in the D3 paper to assess the quality of the reconstructed genomes. This would significantly strengthen your argument.
>
> **Focus on Genotypes**: I understand the focus on genotypes due to their relevance to GWAS and their availability. However, providing a quantitative justification for this choice instead of a qualitative one would make your argument more compelling.
>
> **Literature Review**: I appreciate your willingness to consider adding DNADiffusion.  A comprehensive review of relevant work, even if tangentially related, strengthens the paper's context and contribution.
>
> **Convolution-based U-Net**:  The U-Nets I mentioned all rely on convolutional layers.  Could you please provide a quantitative comparison showing the performance advantage of the selected custom CNN-based U-Net over standard architectures?
>
> **Experimental Section**:  The real-world scenario you describe is understandable. However, the current experiments don't fully convince me that generating synthetic genotypes in this scenario is feasible and effective.  To bridge this gap, consider designing experiments that more closely mimic the scenario you outlined, demonstrating the utility of your method for researchers with limited genotype data.
>
> **Real-World Scenario**: Thank you for elaborating on the real-world application. I still believe the paper needs further clarification and stronger evidence to support the feasibility and effectiveness of your method in this context.
>
> I appreciate that you have addressed some presentation issues, such as replacing screenshots with actual plots. However, I urge you to carefully review my remaining comments, especially regarding the notation in Equation 3, which needs proper introduction and discussion.
>
> **Overall Assessment**
>
> Based on the authors' responses and the current state of the manuscript, I stand by my initial assessment that this paper is not yet ready for publication at ICLR. The technical concerns regarding the evaluation pipeline and the lack of strong empirical evidence remain. The paper presentation also does not meet the expectations of a conference such as ICLR in my opinion.  I am happy to discuss my perspective further with the other reviewers and the area chair.
>
> I believe that generating full genotypes using diffusion models is a promising direction. I encourage you to refine your paper presentation, strengthen the experiments, and address the remaining concerns to prepare a stronger manuscript for a future conference.

---

> > ### Author Response · Authors · 2024-11-26
> >
> > Dear Reviewer,
> >
> > While we appreciate your effort you put into the reply, we have remained somewhat puzzled about how to deal with it.
> >
> > In our opinion, there are issues which are not correct, such as the following:
> >
> > - You now acknowledge the novelty of our efforts, and apparently seem to understand the achievement. On the other hand, you continue to suggest to re-establish full-length genomes from latent space embeddings, and claim that such an analysis would be imperative. However, as we pointed out in our response, such a reconstruction is pointless with respect to the application scenario that we address.
> >
> > Note in addition that it is common practice in GWAS type studies to operate in latent space, because genotype data is routinely PCA'd before conclusions can be derived.
> >
> > What, exactly and in full detail, do the experiments look like that you have in mind?
> >
> > - You suggest to "consider designing experiments that more closely mimic the scenario you outlined, demonstrating the utility of your method for researchers with limited genotype data". However, exactly that we do: we demonstrate that providing researchers holding limited local data with additional synthetic data has the potential to decisively support their analyses. We address your concern one-to-one already.
> >
> > - CNN based U-Nets: We do not understand why we should integrate comparative analyses on possible alternative architectural choices, if the architecture that we crafted works to great advantage, and because minor adaptations to re-designing U-Net architectures is only a very minor aspect of the overall take-home message of the paper.
> >
> > - Further, there are very generic suggestions such as the claim that we have 'an incomplete literature review' and the suggestion to add  'tangentially related work to strengthen the paper‘ - if related work is tangential, it is not relevant.
> >
> > - Real-World Scenario: Your request for "further clarification and stronger evidence to support the feasibility and effectiveness of your method in this context" is very vague. We would appreciate considerably more specific comments such that we can improve the manuscript. We are wondering what kind of 'clarifications' you expect, and what your specific and detailed suggestions for additional experiments are.
> >
> > Therefore we do not see how we can take these suggestions into account.

---

### Official Review · Reviewer_nELF · 2024-11-04

**Soundness:** 3
**Presentation:** 2
**Contribution:** 2
**Rating:** 5
**Confidence:** 3

**Summary:**

The authors develop diffusion models to generate realistic human genotypes. Their claimed contribution is the first models to produce full-length human genotypes. They develop CNN-, CNN+MLP- and Transformer- based models optimized for reconstruction loss. They evaluate on ALS and 1000 Genomes data. They evaluate the ability of the models to augment classifier training and also test the privacy loss associated with generating genotypes with the models.

**Strengths:**

The paper is relatively clear although the language and grammar could use polishing. The claimed contribution is clear and the authors discuss prior work clearly. For the most part the paper is easy to follow. The problem of synthetic data generation is a significant one and the development of synthetic human genomes is surely of interest to some in the research community, although I do not work in this area.

**Weaknesses:**

Although the authors discuss prior work, they do not seem to compare to modeling approaches used in it. It is not clear if this is because of technical limitations or an oversight. The motivation for generating synthetic genotypes to improve ML models is clear, but the authors do not spend enough time discussing the possible negative ethical implications of generating human genotypes. I appreciated the evaluation of privacy loss but the implications were not fleshed out.

Specific comments:

- All of the figure texts are too small to read.

- change " to `` for front quotes

- " that do not only" -> "that not only"

**Questions:**

I would like a more thorough ethical discussion including, but not limited to, privacy concerns. What would a bad actor do with this technology?

**Details Of Ethics Concerns:**

The motivation for generating synthetic genotypes to improve ML models is clear, but the authors do not spend enough time discussing the possible negative ethical implications of generating human genotypes.

---

> ### Author Response · Authors · 2024-11-19
>
> We would like to thank you for your review. Although you say that you do not work in this area, you point exactly to the things that matter, namely ethical concerns and privacy.
>
> In the following, we will address your Questions and the Weaknesses you pointed out, one by one.
>
> Questions:
>
> What would a bad actor do?:
> We will integrate a paragraph that summarizes these ethical aspects into a revised version of our manuscript. We answered below in more detail.
>
> Weaknesses:
>
> On negative ethical implications: We are now providing a more thorough ethical discussion in the revised version of our manuscript, discussing the possibilities of a bad actor.
>
> In particular we go into criteria as also specified in the AI-Act of the EU or the US AI Bills of Rights, namely privacy, security, fairness and robustness.
>
> In general we envisioned the diffusion model being trained in a secure environment and the synthetically generated privatized data being released for further research. To obtain privacy, we suggest mechanisms which lead to provable guarantees, such as differential privacy or multiparty computation. As the transfer of SNP data in between different locations might harm privacy, such methods could be combined with federated learning technologies.
> A further challenge is a possible bias which can occur due to a skewed training set. Evaluation whether biases exist can be based on downstream tasks. Bias mitigation could be based on resampling strategies for the training data.
> Finally, the same as all AI models, we expect that the method is vulnerable to attacks such as data poisoning or adversarial attacks of downstream tasks. Thus, the credibility of data sources and robustness of downstream models needs to be assured. We think, however, that the implementation of these variants is out of the scope of the current paper.
>
> Comparison with other modeling approaches. This is, by no means, an oversight. In fact, the lack of methods against which to compare is the very motivation of our paper. We were aiming to suggest, to the very best of our knowledge, the first approach by which to generate full-length human genotypes. After thoroughly and carefully revisiting the landscape of existing tools and approaches (see Table 1 in the manuscript), we came to conclude that there are indeed no approaches that can generate synthetic full-length human genotypes (or even human genomes directly at the level of DNA). All approaches presented so far address generating smaller segments of human genomes, not even spanning the length of one chromosome. However, classifiers as the one presented in [3] require full-length human genome data to work well.
>
> This situation prevents a meaningful and fair comparison with other approaches, or even renders a comparison technically infeasible. For the last point, consider that one could generate smaller patches of the human genome using the corresponding approaches, and subsequently stitch them together to obtain a full-length genome. However, such a procedure would require training several thousands of generative models and to sample from them. Estimated runtimes are to be measured in terms of months, which prevents such practice for obvious reasons (blockage of resources, explosion of electricity bills, etc).
>
> Minor, specific comments: We have fixed all problems mentioned, thanks for pointing them out.
>
> [1] https://cybersecurity.springeropen.com/articles/10.1186/s42400-021-00105-6
> [2] https://www.ncbi.nlm.nih.gov/gap/
> [3] Xiao Luo, Xiongbin Kang, and Alexander Schonhuth. Predicting the prevalence of complex genetic ¨ diseases from individual genotype profiles using capsule networks. In Nature Machine Intelligence. Nature Publishing, 2023.

---

### Official Review · Reviewer_Q7FW · 2024-11-04

**Soundness:** 3
**Presentation:** 3
**Contribution:** 3
**Rating:** 8
**Confidence:** 3

**Summary:**

The authors present a novel approach to generating full-length synthetic human genotypes with diffusion models. They empirically show that the synthetic genotype closely resembles the actual data. They show that the performance of disease and population classifiers was maintained when synthetically generated data was used for training those classifiers. The study also demonstrates that augmenting the training data with synthetically generated data improves the performance of the classifiers. This approach can significantly improve data availability and access and preserve data privacy.

**Strengths:**

1. The paper is well-written and presents a technically sound and well-structured approach.
2. Using diffusion models to generate full-length genotypes is a novel approach. The rationale and motivation for this approach are clearly stated.
 2. The paper presents a significant contribution to addressing the challenges with data access restrictions and privacy in genome data.
 3. The paper presents empirical evidence for comparable performance between the models trained with synthetic and real data.

**Weaknesses:**

1. Authors should discuss potential limitations of their current evaluation approach in more detail.
2. Could authors discuss any limitations in generalizing their approach to other genomic datasets?

**Questions:**

1. Can the authors explain why they opted for variable PCA component usage and did not use a consistent number, such as 8, for all genes?
2. Given its relevance for assessing distributional similarity, why was the Frechet Distance not used for holdout evaluation?
3. Could authors clarify their data splitting procedure and discuss any potential implications for the validity of their results?

---

> ### Author Response · Authors · 2024-11-19
>
> Thank you very much, we appreciate your constructive feedback.
>
> We will first answer your Questions, and continue by addressing the Weaknesses you pointed out.
>
> Questions:
>
> Variable numbers of PCs: We agree that universal usage of 8 PCs across genes would be a more principled approach, and we can imagine that such unified usage of PCs could also lead to improved results. Nevertheless, we followed [1] in their choices, supported by the rather encouraging results presented there.
>
> Frechet Distance: We agree that in principle, using FID related scores would be desirable, so we also considered using it. However, note that evaluating the FID requires the availability of pre-trained Inception networks. Thereby, Inception networks act like a fully approved choice to support high-performance classification of images. In our case, dealing with genomes / standard genetics data, such canonical, off-the-shelf, high quality classification models do not (yet) exist. So, we resorted to using the metrics described in the manuscript.
>
> Data Splitting: The data splitting procedure used for the 1KG dataset followed standard prescriptions: a random sampled 10% of the dataset was used for evaluation (as test data). For the ALS dataset, analogously, we randomly sampled 10% of the data such that both ALS positive and ALS negative cases were represented in equal amounts (balanced test data). In this, we again aimed to match data splitting schemes suggested in [1].
>
> Weaknesses:
>
> Limitation of evaluation: We agree with you on discussing this in more depth, because this is a great point. The design of metrics by which to evaluate realism and diversity of generated genome data is, while an area of research in its own right, still in its infancy. We have tried to point this out in Section 3.4 where we discuss evaluation methods for image diffusion models and updated it in the revised manuscript to make it clearer. The application of various such methods is technically not possible or does not make sense in our context. So, we employed the evaluation measures used in our paper, drawn from other domains, as measures that are both applicable and make sense. These measures are still popular, although not as frequently applied as, for example, the FID in the context of image generation.
>
> Generalization to other genetic datasets: The method we propose generalizes to any genetic dataset that relates to genotypes. Note that, although we solely focus on SNP sites here, any polymorphic sites (e.g. also relating to larger structural genomic variants) can be integrated into our approach. More generally, any DNA sequence stemming from an organism for which a reference genome is available, can be turned into genotype data, hence becomes suitable for usage in our context. As a proof of the generality of our approach, note that selections of polymorphic sites differed substantially when considering ALS data on the one hand, and the 1000 Genomes (1KG) data on the other hand. Regardless, our approach could be applied without any redesign.
>
> It is not straightforward, however, to merge genotype datasets from different platforms, if polymorphic sites are not identical across datasets. Various ways to mitigate this issue are conceivable, from imputation of missing data to replacing missing data using special symbols are conceivable. All of those are standard in computational genetics, so could be straightforwardly implemented also in our context.
>
> [1] Xiao Luo, Xiongbin Kang, and Alexander Schonhuth. Predicting the prevalence of complex genetic ¨ diseases from individual genotype profiles using capsule networks. In Nature Machine Intelligence. Nature Publishing, 2023.

---

> > ### Comment · Reviewer_Q7FW · 2024-11-26
> >
> > The authors provided appropriate responses that directly addressed the this reviewer's concerns. I updated my review rating accordingly. Thanks

---

### Official Review · Reviewer_Dfcy · 2024-11-04

**Soundness:** 4
**Presentation:** 4
**Contribution:** 4
**Rating:** 10
**Confidence:** 4

**Summary:**

This article presents a method for generating full synthetic human genomes using diffusion. Four diffusion architectures are compared, and the quality of the generated genomes is analyzed. The article demonstrates that this method can generate useful synthetic data for geneticists.

**Strengths:**

This article is very exciting. It is incredibly well-written with a clear and enjoyable prose. It tackles an important issue, human genome analysis, with state of the art methods, diffusion models. It presents an interesting comparative study for deep learning in a data regime not often considered, especially in the diffusion literature, but of great import and scientific interest. The insights from the architecture comparison study are useful for a machine learning readership beyond the biomedical application; an understanding of diffusion methods applied to different data types, and how generated data can be measured and made useful, is relevant to ICLR. The demonstration of generating full human genomes is both an impressive showcase of the capabilities of diffusion and an important step in the study of the human genome.

**Weaknesses:**

It would have been helpful to contextualize the results in comparison to other methods, like the work of Szatkownik et al, or HyenaDNA. It is unclear to me how feasible the application of these methods to the full human genome is, but this could have been discussed in the results. However, it should be possible to generate smaller sequences with the proposed diffusion method in order to compare to previous works. The experimental results, as they are, demonstrate the viability of diffusion models on the full human genome (and provide a useful architecture comparison). However, they do not give an indication of the relative quality of the generated data over the data generated by other works.

Other than that, I identified no major weaknesses. Rather, here are some minor quibbles.

+ the plots are not vectors and have small fonts, making them blurry in standard sized readers. pdf plots with larger fonts would help with clarity.
+ line 451: "We draw two conclusions from this:" this is an incomplete sentence, or the next paragraph shouldn't be presented as a new paragraph.
+ eq 2 could be defined inline
+ lines 453 and lines 483 begin paragraphs that say roughly the same thing. Those two paragraphs could be merged or each trimmed.
+ there's an unneeded comma in line 492

**Questions:**

The MLP architecture seems very shallow, especially compared to the CNN, and the first downscale layer, from 147k to 1024, seems very abrupt. What motivated these choices? Were deeper architectures tried?

What is the explanation for the MLP loss not moving? line 363: "even though only small drops in loss during training can be observed for
the MLP, the reconstruction error keeps improving." Is the MLP learning the data distribution? Is it acting as more than a regularization term in the MLP+CNN architecture?

It seems odd that a CNN, albeit with 1D kernels, would outperform a transformer on such long sequences. What are the intuitions behind the transformer architecture not working as well, especially on recovery rates of data coming from the transformer? This seems odd. I agree with the comments on lines 760/761, but am curious if there was more analysis here.

A greater exploration of DDIM could have been nice. Was DDIM tried, and if so, at what timestep reduction? If DDIM is used just for inference but not for training, is there a large decrease in quality?

**Details Of Ethics Concerns:**

The ethical concerns are addressed in section 5, however there is also a question of data bias which isn't discussed in this section. Is it possible that the diffusion methods create biased synthetic data that misrepresent the real data distribution? The latent space visualizations seem to suggest this, as there are concentrations of synthetic data outside the distributions of real data. Given that the data is human genomes, the ethical implication is that synthetic data generated through this method could over-represent one subset of the original human data and under-represent another, leading to bias in downstream tools that use the synthetic data.

---

> ### Author Response · Authors · 2024-11-19
>
> Thank you very much, we sincerely appreciate your positive review. We will first answer your Questions, and continue by addressing the Weaknesses you pointed out.
>
> Questions:
>
> MLP Architecture: Indeed, the MLP architecture is comparatively shallow. We settled on this architecture after various iterations. We reckon that the shallowness is due to the tendency of MLPs to over-fit at excessive amounts of parameters. Note that the first layer in particular, from 147k down to 1024 neurons already consumes 147.000 x 1.024 = 48.128.000 parameters, which is a huge amount for a single layer. Increasing the number of neurons in the hidden layer, for example to 2048 or 4096 substantially increases the amount of parameters. We also verified the tendency to overfit experimentally, without explicitly mentioning this in the paper.
>
> MLP loss not moving: While only slightly in comparison with other tracked losses, the MLP loss is actually still decreasing, which may not be immediately apparent. We reckon that this is due to the MLP encountering problems when refining the smaller scale structures of the genome (see Figure 5 in the Appendix), because it predominantly focuses on the recovery of the larger, overarching structures.
>
> Clearly, the MLP keeps learning despite only the small reduction of loss, because stopping training early leads to reduced recovery rates (see Table 2 for final recovery rates). Furthermore, the reconstruction error keeps decreasing during training (see Figure 3).
>
> CNN vs Transformer: We agree that this is odd: in general, Transformer based architectures should work for the type of data at hand. Note that for the classification tasks Transformers achieve good performance (85% accuracy vs 87% MLP). This is definitely an interesting question why for data generation on this data specifically transformers seem to struggle.
>
> DDIM: We also experimented with DDIM (see lines 91-92 in the manuscript), using 50 instead of 500 generation steps as was used for DDPM. As DDIM generally produces slightly worse results than a full DDPM (see the original DDIM paper [1], Table 1) we decided to focus on the generative approach yielding highest quality. If, for a particular application, generation speed was a bottleneck, DDIM would of course become an option of potentially greater interest.
>
> Weaknesses:
>
> Major:
>
> We agree that more contextualization would be helpful. However, all other methods can not produce full genomes, which are needed to obtain high performances on classifiers which need whole genome data. I.e. ALS classification. In principle we could of course stitch together the small segments generated by other methods. This would however require a high amount of compute as well as manual effort and is thus out of the scope of this paper.
>
> Minor points:
>
> We will address the minor points in the camera ready version, many thanks for pointing them out.
>
> Ethics Statement:
> Thank you for pointing this out. In Lines 442-446 we also discuss that the data generated by the MLP model seems to lie out of distribution of the real data i.e. the generated data is biased. For the other model types we do not observe this. That U-Map does not visualise this is of course not proof that no bias exists. In principle we think that the more data we add to the training process of our diffusion model the better it will generalise and the less biased the generated data will be. This study was done on comparatively little training data (~10000 datapoints). For real life applications we hope that a larger and well balanced training set will mitigate this risk, but further studies need to be employed to gain more understanding.
>
> [1]Jiaming Song, Chenlin Meng, and Stefano Ermon. Denoising diffusion implicit models, 2022.

---

### Meta-Review · Area_Chair_u45v · 2025-01-12

**Metareview:**

This paper presents a diffusion based approach to generating synthetic genotype data. The idea is interesting and presents a new direction for the field that merits further exploration. However, as pointed out by reviewers iupA and ydeS, the current manuscript suffers from many issues, most notably immature experimental evaluation that hints at potential utility but does not unambiguously demonstrate it. The paper in its current form is more a proof-of-principle, and an imperfect one at that.

**Additional Comments On Reviewer Discussion:**

Reviewer ydeS comment that:

“My biggest question concerns how exactly you augment the data. There are no details about how you utilize the conditional generating capabilities. Given a small amount of data, how do you select the conditions? What is the relationship between your "seed" training data and the generated data?”

gets at the heart of the problem. How synthetic data is combined with real data to produce the results in Table 3 is unclear. But, reading between the lines of the authors’ response, it would appear that conditional data from the synthetic set are used to augment the real data. E.g., both classes of ALS are used from the synthetic pool, which strongly undermines the utility of this method, because it implies that the synthetic generator must have had access to large amounts of relevant labelled data. The authors’s proposed use case, of using dbGap data as a general form of augmentation, would not be applicable here as data on the target phenotype to be predicted needs to be available in dbGap. And at any rate, this sort of scenario ought to be actually demonstrated instead of only mentioned as a hypothetical possibility.

---

### Decision · Program_Chairs · 2025-01-22

Reject